# The Impact of Electronic Nicotine Delivery System (ENDS) Flavors on Nicotinic Acetylcholine Receptors and Nicotine Addiction-Related Behaviors

**DOI:** 10.3390/molecules25184223

**Published:** 2020-09-15

**Authors:** Skylar Y. Cooper, Brandon J. Henderson

**Affiliations:** Department of Biomedical Sciences, Joan C. Edwards School of Medicine, Marshall University, Huntington, WV 25703, USA; cooper394@live.marshall.edu

**Keywords:** nicotinic acetylcholine receptors, e-cigarettes, vaping, electronic nicotine delivery systems, flavorants, chemical flavors, terpene, menthol, green apple

## Abstract

Over the past two decades, combustible cigarette smoking has slowly declined by nearly 11% in America; however, the use of electronic cigarettes has increased tremendously, including among adolescents. While nicotine is the main addictive component of tobacco products and a primary concern in electronic cigarettes, this is not the only constituent of concern. There is a growing market of flavored products and a growing use of zero-nicotine e-liquids among electronic cigarette users. Accordingly, there are few studies that examine the impact of flavors on health and behavior. Menthol has been studied most extensively due to its lone exception in combustible cigarettes. Thus, there is a broad understanding of the neurobiological effects that menthol plus nicotine has on the brain including enhancing nicotine reward, altering nicotinic acetylcholine receptor number and function, and altering midbrain neuron excitability. Although flavors other than menthol were banned from combustible cigarettes, over 15,000 flavorants are available for use in electronic cigarettes. This review seeks to summarize the current knowledge on nicotine addiction and the various brain regions and nicotinic acetylcholine receptor subtypes involved, as well as describe the most recent findings regarding menthol and green apple flavorants, and their roles in nicotine addiction and vaping-related behaviors.

## 1. Introduction

Cigarette smoking remains the leading cause of preventable disease and death worldwide, with nearly half a million deaths per year in the United States alone [1]. Additionally, more than 16 million Americans are suffering from a smoking-related disease including diabetes, stroke, cardiovascular disease, chronic obstructive pulmonary disease, or cancer [1]. Nearly 70% of adult smokers in the United States have a desire to quit, however, only 7% are successful [2], with an average of 10 cessation attempts needed for success [3].

In 2009, the Family Smoking Prevention and Tobacco Control Act was put in place to combat adolescent tobacco use and limit cigarette sales by banning all flavor additives other than menthol from being added to combustible cigarettes, yet this refrained from addressing other tobacco products including cigars, chewing tobacco, hookah, and more. As of 2018, mentholated cigarettes made up 36% of all cigarette sales in the United States [4], however, it has been established that regardless of labeling, even non-mentholated cigarettes contain traces of menthol [5].

Over the past two decades in the United States, combustible cigarette use has declined by nearly 11%, however, one form of nicotine administration has been replaced by another—electronic nicotine delivery systems (ENDS) [6]. ENDS, or electronic cigarettes (e-cigarettes), are handheld devices that vaporize an e-liquid solution, commonly containing varying ratios of propylene glycol and vegetable glycerin, flavoring chemicals, nicotine, and in some cases additional sweeteners [7,8,9]. ENDS were initially intended to be a smoking cessation aid; however, ENDS companies have begun to target a new market of nicotine users among the adolescent population. According to the National Youth Tobacco Survey, usage rates continue to rise with over four million high school students and one million middle school students currently using ENDS products [10,11]. Nicotine dependence is thought to be intensified among adolescents when flavorants are present, as they mask the aversive sensations associated with nicotine and may promote pleasure on their own. This occurs primarily through masking the initial harshness of nicotine/tobacco that is aversive to new and beginning smokers and therefore increases smoking initiation [12,13]. Although non-menthol flavors are banned in combustible cigarettes, >15,000 flavor options are available for ENDS products with a 67% increase in flavor production from 2013 to 2014 [14,15]. In 2019, more than 50% and 60% of high school users used menthol and fruit flavored ENDS, respectively, and more than half of ENDS users prefer flavored products [10,11]. This has become a cause for concern with the number of adolescent ENDS users continuing to rise and the growing popularity of zero-nicotine flavored e-liquids. Yet, little is known regarding the effects of flavors on nicotine dependence and vaping-related behaviors.

Despite this gap in knowledge, there are numerous reports of menthol’s effect on nicotine addiction, including menthol’s ability to enhance nicotine reward and reinforcement [16,17,18,19]. These effects are due to menthol-induced nicotinic acetylcholine receptor (nAChR) upregulation [18,20,21], enhanced dopamine neuron excitability and dopamine release [22,23], and TrpM8-dependent mechanisms [24]. Based on a recent study reporting green apple and other fruity flavors to be the most popular of ENDS flavorants [9,25], additional reports have identified popular green apple flavorants, farnesol and farnesene, to not only enhance nicotine reward in a mouse model, but also display rewarding properties in the absence of nicotine [19,26,27]. These behavioral effects were found to be caused by changes in nAChR upregulation or stoichiometry, and ventral tegmental area dopamine neuron firing. Based on these findings, it is critical we further understand how ENDS flavoring chemicals may alter the addictive properties of nicotine in an attempt to combat the growing ENDS-use epidemic.

This review summarizes the current knowledge base of nicotine addiction and the major neurobiological and neurophysiological adaptations that contribute to dependence. Additionally, we summarize the effects of menthol on abuse liability and vaping-related behaviors and include the major impacts that another popular ENDS flavor, green apple, has on addiction-related behavior. This includes their effects on nicotine’s actions in the brain and the major neurocircuitry involved in the induction of addiction.

## 2. Background of Nicotine Addiction

### 2.1. Neuronal nAChRs: Structure and Function

nAChRs are ligand-gated ion channels in the Cys-loop superfamily, alongside N1-type acetylcholine, GABA_A_ (ionotropic form), glycine, and 5-HT_3_ receptors [28,29]. nAChRs are responsible for mediating fast synaptic transmission of nerve impulses [30,31,32,33]. Human nAChRs are assembled from various combinations of subunits, including α2–α7 and β2–β4. Assembled subtypes consist of homomeric pentamers (α7) or heteromeric pentamers (α2–α6 with β2–β4; Figure 1A). Each subunit comprises of a large *N*-terminal extracellular domain important for ligand binding, a short *C*-terminal extracellular domain, four hydrophobic transmembrane domains (M1–M4), with M2 lining the channel’s central pore, and a large cytoplasmic loop between M3 and M4 that varies among different subunit complexes [34] (Figure 1B,C). The cytoplasmic loop is important for receptor trafficking in both the anterograde and retrograde directions between the endoplasmic reticulum to the plasma membrane [34,35,36,37]. These sites are also critical for protein interactions and serve as a site for phosphorylation.

nAChR agonists include the endogenous neurotransmitter, acetylcholine (ACh), as well as various exogenous molecules. Homomeric receptors have five identical orthosteric ACh-binding sites, whereas heteromeric receptors contain two or more orthosteric sites at the interfaces between an α and β subunit (Figure 1A). The binding of two ACh molecules to the orthosteric sites on the receptors induces a conformational change allowing the channel to open. nAChRs are permeable to monovalent Na^+^ and K^+^ ions, as well as divalent Ca^2+^ ions. Permeability to calcium acts on intracellular cascades that can play a vital role in neuronal signaling and plasticity [38]. Depending on the subtype assembly or stoichiometry of nAChRs, Ca^2+^ permeability varies. For instance, among α4β2 nAChRs, the high-sensitivity (α4_(2)_β2_(3)_) and low-sensitivity (α4_(3)_β2_(2)_) nAChR stoichiometries exhibit different Ca^2+^ permeabilities [39]; but homomeric α7 nAChRs exhibit the highest Ca^2+^ permeability of all nAChR subtypes [40,41]. The net flow of positive ions inward depolarizes the cell membrane causing an excitatory postsynaptic potential [30]. nAChRs have a very widespread distribution throughout the brain and are found on presynaptic and postsynaptic neuronal membranes, as well as non-neuronal cells, such as glial cells [42,43]. nAChRs can control excitatory and inhibitory neurotransmission, which further excites and/or inhibits target cells [30,44]. Depending on the brain region and cell type, nAChR subunits form differing complexes that are associated with a variety of pathophysiological conditions. The most widely expressed neuronal subtypes in the brain are heteromeric α4β2* (* = nAChR may contain other subunits) and homomeric α7 nAChRs [45,46,47,48]. Due to the variability in nAChR formation, each complex differs functionally with respect to channel opening, closing, and desensitization [47].

### 2.2. Nicotine’s Actions in the Brain

Nicotine crosses the blood brain barrier and binds with high affinity to nAChRs widely distributed throughout the nervous system [32,38,48,49,50]. This interaction promotes a variety of neurophysiological changes including chaperone-mediated nAChR upregulation [30,35,37,51,52,53,54], activation of the mesocorticolimbic reward and reinforcement pathways [55,56], enhanced synaptic plasticity [57,58,59,60], and enhanced neuronal firing [53,60,61,62,63], ultimately leading to the development of nicotine addiction [33,64] (Table 1). This will be expounded in detail in the following paragraphs.

Nicotine binding promotes a conformational transition of nAChRs from a resting, closed state to an open state, allowing signal transduction to occur [65,66]. Initially, high affinity and high-sensitivity α4β2* and α4α6β2* nAChRs found on ventral tegmental area (VTA) dopamine (DA) neurons are quickly activated by low nicotine concentrations upon arrival into the midbrain region [67,68,69,70]. Further, nicotine acts on α7 nAChRs on glutamate neurons of local and distal regions resulting in a net excitatory effect on VTA DA neurons and synaptic strengthening between the two neuronal populations [47,60,71]. In addition to this excitatory transmission through DA and glutamate neurons, activation of the α4β2* nAChRs found on GABA neurons of the midbrain elicits an inhibitory tone to VTA DA neurons, reducing the transmission of DA and decreasing reward through the mesolimbic pathway (dopaminergic tracts from the VTA to the nucleus accumbens (NAc)) [72]. However, the activation of α4β2* and α7 nAChRs on midbrain DA and glutamate neurons, respectively, promotes a net excitatory effect on DA neurotransmission from the VTA to the NAc and prefrontal cortex (PFC), leading to the rewarding and reinforcing aspects of nicotine use [71]. More recently, Yan et al. discovered functional β2* nAChRs on medial VTA (mVTA) glutamate neurons that further contribute to this net effect [63,73]. In general, nicotine exerts strong effects on the local microcircuits of the VTA, which have differing acute and chronic effects. This will be discussed further in a later section.

Persistent nicotine exposure further transitions the receptor to a desensitized state, where it is less responsive to agonist stimulation [34,44,45]. nAChRs are activated and desensitized to a degree depending on the subtype, brain region, and concentration of nicotine. According to Brody and colleagues, nAChRs will activate and desensitize at brain nicotine concentrations of 20–100 nM following cigarette smoking [74]. Following activation, most α4β2* nAChRs desensitize, which decreases the GABAergic transmission onto DA neurons resulting in the disinhibition of DA neurons (Figure 2C). These effects are reinforced by the enhanced glutamate neurotransmission from distal regions and local mVTA glutamate neurons to lateral VTA (latVTA) DA neurons [63,73]. This depolarization and enhanced action potential firing of glutamatergic neurons leads to long-term potentiation (LTP) and synaptic strength onto the midbrain DA neurons, which is a key role in the formation of nicotine dependence [53] (Figure 2). Additionally, although the α4β2* nAChRs present on VTA DA neurons desensitize similar to VTA GABA neurons, these DA neurons also express α6β2β3* and α4α6β2* receptors, which do not desensitize as quickly as the high affinity α4β2* nAChRs [55]. Overall, there continues to be a net excitatory effect on DA release in the presence of nicotine molecules.

When an individual becomes a long-term nicotine user, further neuroadaptations occur. A major hallmark of nicotine dependence has been the occurrence of nAChR upregulation. This phenomenon has been understood by many to be a post-translational occurrence, based on no observable changes in mRNA levels [35,75,76]. Similar to desensitization, upregulation differs in response to nicotine concentration and time course [77], and is brain-region, cell-type, and nAChR subtype-specific, given that no upregulation has been noted in the thalamus, high-affinity nAChRs are favored for plasma upregulation, and only α4, α6, β2, and β3 subunits upregulate. Upregulation occurs through a process termed pharmacological chaperoning that involves intracellular actions of nicotine that promote nAChR subunit assembly and enhanced trafficking of nAChRs through the secretory pathway [35,37,52,76]. This process is a physiological response following nAChR desensitization [45,78] that takes approximately ten days of long-term use among human and rodents but can also occur in cultured neuron and cell line preps as well [76,79]. Through the use of fluorescently tagged nAChRs, changes in nAChR number have been measured without the use of radioligand binding assays [18,20,35,37,53]. As previously stated, high-affinity α4β2 nAChRs are stabilized during this process [37,52,80]. With higher nicotine concentrations, further nAChRs are subjected to upregulation as well.

Given these effects are based on freebase nicotine that is present in combustible cigarettes, it is critical to point out the distinct differences between the combustible cigarette and the electronic cigarette that may affect the timespan of these steps towards addiction. With JUUL (pod-based e-cigarette) being one of the most popular ENDS devices among the adolescent population, studies have begun to surface identifying what makes JUUL so popular. The major difference between these products is the use of nicotine salt in JUULs [81]. Nicotine salt differs from nicotine freebase in that it contains benzoic acid in order to sufficiently protonate most of the nicotine. The use of nicotine salt results in faster absorption, increased nicotine strength, and can be palatable at high concentrations, as opposed to nicotine freebase. Based on the different pharmacokinetic properties of these two nicotine forms, JUUL’s average nicotine concentration per e-liquid pod is ~60 mg/mL, or 1.5 times the nicotine concentration in a pack of cigarettes. Although combustible cigarettes range in nicotine concentration, the majority contain 2 mg of nicotine per cigarette. Additionally, various other ENDS products (tank-based e-cigarette) also exhibit higher nicotine concentrations than the average cigarette (3–18 mg/mL), however, they are still much lower than the prefilled JUUL pods. Based on these differences, it is likely that ENDS enhance the steps of nicotine addiction and are further detrimental to the adolescent brain.

### 2.3. nAChR Subtypes

#### 2.3.1. α4β2* nAChRs

To date, α4β2* nAChRs have been the most-studied subtype in relation to nicotine addiction [33,37,45,55,61,82], as demonstrated through genetically modified mouse models [55,56,82,83]. Using knockout mice, Picciotto et al. investigated α4 and β2 subunits and found attenuated nicotine self-administration and conditioned place preference in the absence of these subunits [83]. However, when α4 or β2 subunits were re-expressed in the VTA, a prominent brain region associated with drug reward, it led to recovery of nicotine reinforcement and reward, demonstrating the importance of α4β2* nAChRs in the formation of nicotine dependence, specifically in the VTA. Alternatively, Tapper et al. genetically altered mice to express hypersensitive α4* nAChRs [55]. This approach looked beyond the necessary presence of α4β2* nAChRs in addiction, and instead investigated the specific role(s) that α4 may play in addiction. The presence of hypersensitive α4* not only enhanced the nicotine reward at low concentrations, it also mediated nicotine-induced locomotion through α4* nAChR sensitization, and facilitated tolerance to nicotine, deeming its importance in the induction and maintenance of nicotine dependence. Further, the administration of the β2* receptor-specific antagonist, dihydro-β-erythroidine (DhβE) hydrobromide, into the VTA decreases nicotine reward-related behavior as well as blocks the reinforcing effects of the drug [84,85].

α4β2* nAChRs exist in a high- and low-sensitivity stoichiometry, α4_(2)_β2_(3)_ and α4_(3)_β2_(2)_, respectively [39,86]. Due to its high sensitivity stoichiometry, α4β2* receptors have a high affinity for low concentrations of nicotine commonly present following cigarette smoke inhalation [33,67,87]. Nicotine-induced upregulation of α4β2* nAChRs has been shown to depend on the specific subunit composition, with the high-sensitivity stoichiometry to be favored [35,37,80,88]. Further, it is important to note the more recent identification of genetic etiologies of CHRNA4, CHRNB2, and CHRNB4. Liu et al. positively correlated these genes with smoking phenotypes including ‘age of smoking initiation’, ‘cigarettes per day’, and ‘smoking cessation’ [89].

#### 2.3.2. α6β2β3* nAChRs

Given the knowledge that α6* nAChRs are expressed in dopaminergic neurons [90], initial studies aimed to identify the specific brain regions expressing these subtypes. The expression of α6β2β3* nAChRs is mostly limited to reward-related brain regions, including the DA neurons of the mesolimbic tract, making them critical factors in the induction of addiction [91,92,93,94]. However, investigations into the role of α6* nAChRs in addiction have been very limited, with one of the first critical findings being reported by Pons et al. [56]. Utilizing an intravenous nicotine self-administration paradigm and α6 knock-out mice, Pons et al. discovered the essential role of α6* nAChRs in nicotine reinforcement by demonstrating the lack of self-administration behaviors in α6 knock-out mice compared to their wildtype counterparts [56]. Furthermore, re-expression of α6 through a lentiviral vector into the VTA significantly increased self-administration behaviors to a similar extent as the wildtype control mice. Nicotine-induced activation of the mesolimbic system is dependent upon the presence of α4β2* and α6β2β3* nAChRs, as seen through conditioned place preference and self-administration assays [56,83,95]. Similar to α4β2* nAChRs, α6* nAChRs consist of two stoichiometric forms, α6β2 or α6_(2)_β2_(2)_β3 [20,35]. The inclusion of the β3 subunit results in a much higher sensitivity to nicotine. Further, α6β2* nAChRs desensitize to a similar extent as α4β2* nAChRs, however α6β2* nAChRs recover more quickly [96] and are thus more amenable to nicotine’s long-term effects.

#### 2.3.3. α4α6β2* nAChRs

Despite the wide presence of nAChRs throughout the central nervous system, recent studies point to α4α6β2* nAChRs as a principle nAChR subtype that mediates nicotine reward. As discussed earlier, deletions of α4 have shown the importance of this subunit; but additional studies with mouse models tell us that α4α6β2* nAChRs mediate the rewarding effects of nicotine. Deletion of α4, α6, or β2 nAChR subunits is sufficient to block the self-administration of nicotine in mice (Pons et al., 2008). The selective re-expression of these deleted subunits in the VTA was sufficient to reinstate self-administration of nicotine (Pons et al., 2008). Re-expression of these subunits in neighboring midbrain regions, the substantia nigra pars reticulata (SNr) and substantia nigra pars compacta (SNc), did not ‘rescue’ nicotine reinforcement. The nicotine-induced enhancement of DA neuron excitability that is required for reward is dependent upon α4α6β2* nAChRs [69,70]. Here, both Liu et al. and Engle et al. revealed that the presence of both α4 and α6 were necessary for smoking-relevant concentrations of nicotine (300 nM) to depolarize nAChRs on VTA DA neurons. This suggests that while studies revealing the importance of α4 and α6 nAChR subunits independently, the physiological response to smoking-relevant concentrations of nicotine (≤300 nM) depend on α4α6-containing nAChRs.

#### 2.3.4. α7 nAChRs

Alongside α4β2*, α6β2*, and α4α6β2* nAChRs, the α7 nAChR subtype also plays a prominent role in the formation of nicotine dependence. This homomeric nAChR subtype is largely present on glutamatergic neurons in the prefrontal cortex (PFC), hippocampus, and mVTA. Expression of α7 nAChRs in these brain regions play multiple roles in nicotine addiction; they aid in the formation of functional synapses [49], promote the net excitatory effect on midbrain DA neurons of the reward pathway [60], and enhance postsynaptic excitation through NMDA receptors in the hippocampus and prefrontal cortex [49,60]. More recently, a population of functional heteromeric nAChRs has been discovered on mVTA glutamatergic neurons [63]. This novel finding supports the previous work demonstrating the intricate connections between VTA glutamate and DA neurons in reward and reinforcement processing [60,71,97], but further expands this knowledge by observing that nicotine mediates the excitatory transmission at this connection and thus, amplifies the mesocorticolimbic reward transmission [63].

The α7 nAChR exhibits different receptor kinetics compared to α4β2* nAChRs, exemplified by fast activation and desensitization [44]. Due to their desensitization properties, α7 nAChRs recover very rapidly, demonstrating the net excitation discussed earlier. The net excitation through rapid recovery following desensitization, and high permeability to calcium, makes α7 nAChRs critical factors in the induction of long-term potentiation (LTP) [46,98]. Nicotine-induced synaptic plasticity, or LTP, of glutamatergic neurons in the VTA, PFC, and hippocampus are important in the tolerance and associative learning of nicotine addiction. Regardless of the fast depolarization of VTA DA neurons through nicotine’s effect on α4β2*, α6β2*, and α4α6β2* nAChRs, these receptors desensitize fairly quickly and recover slowly, rendering them inactive to the remaining nicotine molecules circulating in the brain. Thus, it is the presence of α7 nAChRs and their rapid recovery that contribute to the long-standing effects of long-term nicotine. Additionally, administration of methyllycaconitine (MLA), an α7-selective antagonist, did not alter nicotine CPP, but decreased self-administration, confirming α7 nAChRs’ role in reinforcement but not reward [95]. These findings were further verified by the lack of change in nicotine self-administration behavior between α7-KO and α7-WT mice [56].

It is the reinforcing properties of nicotine that promote dependence, despite the many health risks. VTA DA neurons receive local and distal (mainly PFC) excitatory glutamatergic inputs that further prompts DA release into the NAc, amygdala, and PFC, which makes up the mesocorticolimbic system [99]. However, when nicotine is persistently present in this system, it causes modification of neural circuits that promotes drug-induced synaptic plasticity [100]. This phenomenon outlasts the effects of the presence of nicotine molecules and contributes to the formation of tolerance and memory consolidation to nicotine. It is the repetitive use and reinforcing properties of nicotine that causes the brain to form an association between nicotine use and the physiological response to nicotine use. This association is what drives nicotine dependence by mediating the behavioral effects of nicotine addiction, including cue-induced craving and reinstatement (in rodents) or relapse (in humans) [101].

#### 2.3.5. α3* and α5* nAChRs

Genetic polymorphisms in the α3/α5/β4 nAChR subunit gene cluster has been linked to increased risk to nicotine addiction [102,103]. Variation in the amino acid sequence of CHRNA5 results in reduced α5* nAChR activity and increases the risk for dependence [102,104,105]. Additionally, Fowler et al. identified a null mutation in the CHRNA5 gene that results in significantly more intravenous self-administration nicotine consumption, compared to their control mice [106]. Interestingly, they found that α5* nAChRs are involved in mediating aversive stimuli, titrating nicotine intake, and mediating somatic signs of withdrawal.

The α5* and α3β4* nAChRs are mostly expressed in the habenulo-IPN pathway, a group of cholinergic and glutamatergic neurons that project from the medial habenula (MHb) to the interpeduncular nucleus (IPN) through a bundle of axons termed the fasciculus retroflexus [33,106,107]. Due to their low affinity for nicotine, these subtypes require high concentrations of nicotine to sensitize them (2 mg/kg or more) [47,106,107]. The opening of α5* or α3β4* nAChRs activates the aversive pathways involved in the mediation of nicotine intake. The activation of either the habenulo-IPN pathway or the lateral habenula (LHb) projections to the rostromedial tegmental nucleus (RMTg) and VTA is a response to the aversive aspects of nicotine, including high doses of nicotine or nicotine withdrawal (discussed later).

## 3. Neurocircuitry Involved in Nicotine Addiction

### 3.1. Ventral Tegmental Area (VTA)

The ventral tegmental area is among the most vital brain regions involved in addiction. The VTA is a heterogeneous collection of neurons that in part makes up the midbrain. Although primarily studied for its DA neurons, the VTA also consists of γ-amino butyric acid (GABA) and glutamatergic neurons [108]. VTA DA neurons have been widely studied in various drug-related experiments for nearly all drugs of abuse acts on VTA DA neurons to stimulate DA neurotransmission into the ventral striatum, or nucleus accumbens (NAc)—commonly known as the mesolimbic pathway [68,109,110,111]. Further, nicotine-induced activation of this pathway stimulates reward, as seen in conditioned place preference (CPP) assays [26,55,82,112]. Whereas, lesioning of this pathway completely abolishes nicotine reward-related behavior, nicotine self-administration, and nicotine-induced locomotion [85,113,114]. Additionally, VTA DA neurons project to the prefrontal cortex (PFC) through the mesocortical pathway, and synapse on cortical pyramidal neurons and GABAergic interneurons. This dopaminergic pathway is largely associated with drug reinforcement and is commonly activated during nicotine self-administration studies [115,116]. Together, these paths make up the mesocorticolimbic DA pathway and are often linked in studies on drugs of abuse, although there are additional projections to various other brain regions, including the hippocampus and amygdala. Each of these neuronal paths circle back to the VTA and mediate the activity of the dopaminergic neurons (discussed further in their corresponding sections).

The VTA also receives afferent projections from numerous parts of the brain that mediate reward or aversion of certain stimuli. The cholinergic and glutamatergic projections of the laterodorsal tegmental nucleus (LDTg) and pedunculopontine tegmental nucleus (PPTg) excite the DA neurons of the VTA leading to burst firing, enhanced dopamine release, and reward-related behavior (Figure 3) [117,118,119]. During long-term nicotine exposure, these glutamatergic projections can initiate long-term potentiation (LTP) in the VTA, promoting a continuous excitatory effect [120]. On the contrary, the medial (MHb) and lateral habenula (LHb) neurons innervate the interpeduncular nucleus (IPN) and rostromedial tegmental nucleus (RMTg) GABAergic neurons, respectively, which employ an inhibitory tone onto VTA DA neurons of the mesolimbic pathway (Figure 3) [121,122]. This net inhibition is stimulated during aversive stimuli, such as high nicotine concentrations, or during withdrawal [106,107,123]. Additionally, the LHb sends direct projections to the VTA in response to aversive stimuli, including the absence of an expected reward [124]. These different afferent projections mediate the firing patterns of VTA DA neurons and influence behavioral outputs.

Although many drugs of abuse studies have focused on DA neurotransmission, GABAergic and glutamatergic neurons of the VTA are prominent factors in the development of nicotine dependence. Opioids, cannabinoids, and benzodiazepines primarily target the VTA GABAergic interneurons, inhibiting their activity and causing disinhibition and burst firing of the neighboring DA neurons [125]. Both GABAergic and glutamatergic interneurons of the VTA send projections to their neighboring DA neurons in order to maintain a homeostatic balance of neuronal firing. Following a cigarette or ENDS puff, nicotine activates the nAChRs found on each of these neuronal populations and increases neuronal firing [126]. However, in a long-term user, these neuronal firing patterns become altered following nAChR desensitization (see Figure 2). The α7 nAChRs expressed on glutamatergic neurons desensitize (and recover) the fastest while β2* nAChRs (α4β2*, α6β2*, and α4α6β2*) desensitize (and recover) much slower. As stated previously, a long-term nicotine user succumbs to various neuroadaptations, including nAChR upregulation and altered neuronal firing. Throughout abstinent periods, the GABAergic neuron firing patterns are enhanced, resulting in a net inhibitory effect on the DA neurons. This not only reduces baseline DA neurotransmission but also causes a depressive state that often triggers craving and relapse. In order to relieve this, the user will consume more nicotine, activate the nAChRs on the various neuronal populations, and increase firing and DA release. With repeated exposure and stimulation, nicotine renders many of the nAChRs inactive. Based on their desensitization properties, the GABA neurons (containing α4β2*) have reduced output, resulting in disinhibition of the VTA DA neurons [71]. Additionally, the fast recovery of α7 nAChRs promotes a net excitatory effect on these DA neurons. It has long been considered that nicotine exerts its effects on α7 and β2* nAChRs on VTA glutamate and GABA neurons, respectively, resulting in a mediatory action on VTA DA neurons. However, recent studies have shown that these glutamate neurons, found primarily in the mVTA, express α4, α6, and β2 nAChR subunits in a somatodendritic manner [73]. Further, Yan et al. has discovered that of the three major neuronal populations in the mVTA (VGluT2^+^ (glutamatergic neurons), Gad2^+^ (GABAergic neurons), and VGluT2^+^/Gad2^+^ (glutamatergic/GABAergic co-releasing neurons)), nicotine enhanced glutamate release into the latVTA via VGluT2^+^ neurons but decreased glutamate (and GABA) release via Gad2^+^ and VGluT2^+^/Gad2^+^ neurons [73]. This further demonstrates the complexity of the VTA, and the various microcircuits involved in reward processing. It is due to these changes that a person dependent on nicotine requires more nicotine over time to achieve the rewarding feeling they once felt. Additionally, more studies are identifying the role of glycine in this microcircuit. Although little work has been performed on glycine’s impact on nicotine addiction, it has been shown that glycinergic neurotransmission significantly alters ethanol intake most likely by impacting glutamate release via glycine receptors on glutamatergic terminals of the VTA [127], however these receptors are also present on GABAergic terminals and have resulted in reduced GABA release and enhanced dopamine release [128,129]. Accordingly, glycine receptors may play a role in nicotine’s actions; but this has yet to be determined.

### 3.2. Substantia Nigra (SN)

The other contributing cluster of neurons that makes up the midbrain is the substantia nigra. The SN is composed of two subgroups: pars compacta (SNc) and pars reticulata (SNr). The SNc is composed of dopaminergic, glutamatergic, and GABAergic neurons while the SNr is largely made up of GABA neurons. The third most extensively studied dopaminergic pathway in the brain aside from the mesocorticolimbic paths is the nigrostriatal pathway, a dopaminergic tract from the SNc to the dorsal striatum [130]. Activation of this pathway is involved in the motor loop of the basal ganglia and has been shown to be acted on by drugs of abuse [131]. The loss of dopaminergic neurons in this pathway is a pathological classification of Parkinson’s disease, characterized by tremors and motor deficits.

### 3.3. Nucleus Accumbens (NAc)

A major output target of VTA DA neurons is the ventral striatum, or the nucleus accumbens (NAc), which makes up the mesolimbic pathway. This pathway is highly involved in reward-related effects of drugs of abuse [68,109]. The NAc contains two subregions: the core and the shell, and is made up of specialized GABAergic neurons, termed medium spiny neurons (MSNs). MSNs consist of two types: D1 (direct pathway) and D2 (indirect pathway) based on the DA receptors present, further explained by Cooper et al. [132]. These neurons receive excitatory projections from the PFC, hippocampus (HIPP), and the VTA [100] to mediate the reinforcing and drug-seeking behaviors of nicotine addiction [133,134]. Repeated activation of the afferent glutamatergic projections can lead to LTP in the NAc, further driving drug-taking behaviors, as demonstrated through reduced AMPAR and AMPAR/NMDAR ratios on MSNs following drug consumption [100]. The D1 MSNs of the lateral NAc shell project in a cyclical manner back to the VTA and synapse on VTA GABA neurons [100,135]. These projections further mediate and disinhibit the VTA DA neuron activity, resulting in enhanced DA release [136,137].

### 3.4. Prefrontal Cortex (PFC)

Dopaminergic projections from the VTA to the PFC make up the mesocortical pathway involved in reinforcement and motivational salience [115,132,133]. This pathway has enhanced neurotransmission during nicotine self-administration, demonstrated by an increase in postsynaptic ionotropic glutamate receptors in the PFC [138]. PFC lesions significantly reduced nicotine self-administration in rats, however, to date, this has only been performed in neonates [139]. Additionally, glutamatergic pyramidal neurons of the PFC project back to the VTA, amygdala, hippocampus, and NAc. The connection between the PFC and NAc is stimulated during drug seeking and reinstatement [140,141]. Due et al. has demonstrated PFC activation of smoker’s brains via functional magnetic resonance imaging in the presence of a smoking-related cue (an image of a person smoking) [142]. Activation of the PFC to VTA pathway results in enhanced firing of the VTA DA neurons and increases DA release through the mesolimbic pathway [143], in part due to upregulation of NMDA and AMPA receptor activation [60].

Based on the growing number of adolescent ENDS users, it is important to discuss the effects of nicotine exposure on the developing brain. More specifically, the PFC is one of the last brain regions to fully reach maturation in the developing brain, which indicates the executive control functions, including attention and working memory are not fully developed upon initial nicotine exposure. Nicotine use or exposure during this critical development period is known to negatively impact the developing process, resulting in impaired cognition and psychiatric disorders, including depression [144,145], and also often results in an increased risk for drug abuse behaviors [146].

### 3.5. Hippocampus (HIPP)

The hippocampus (HIPP) is highly involved in learning and memory, which is of importance to the tolerance-related aspect of nicotine addiction. The glutamatergic neurons of the HIPP are simultaneously excited by VTA DA and PFC pyramidal neurons, which induces LTP in the hippocampus. The strengthening of hippocampal synapses is a key feature to nicotine addiction because of its underlying mechanism involved in cue-induced drug-seeking behavior often leading to relapse [147]. The changes in synaptic machinery are due to the initial formation of immature glutamatergic synapses from the HIPP onto NAc MSNs, leading to silent synapses, which promotes no electrical changes on the inhibitory neurons of the NAc. These immature synapses are due to a large NMDA to AMPA ratio, however, during nicotine withdrawal, stimulation of active synapses occur in part due to the insertion of AMPA receptors on the presynaptic glutamate neurons. This alteration leads to the propagation of inhibitory signals through the reward pathway, ultimately driving drug craving behaviors [148].

### 3.6. Habenula (Hb)

Although most drugs of abuse studies focus on reward pathways, aversive pathways play an important role in addiction as well. The habenula consists of a medial (MHb) and lateral (LHb) portion that has been known to be implemented during nicotine aversion. The most widely studied aversive pathway is the habenulo-IPN tract, which runs from the MHb to the interpeduncular nucleus. The excitatory glutamatergic and cholinergic projections of the MHb activate the GABAergic neurons of the IPN, which then inhibits the DA neurons of the VTA [149]. This, in turn, leads to decreased activation of the mesolimbic pathway and thus, reduced reward.

Although many studies have shown the rewarding and reinforcing aspects of nicotine that drive the drug-seeking and taking behaviors, nicotine is known to have an inverted U-shaped dose response curve in rodents as well as humans [64]. Due to this dose-dependent effect, nicotine tends to be aversive at high concentrations. Humans have been shown to titrate the amount of nicotine they consume when smoking or vaping because of the aversive properties of too much nicotine. The process of titration occurs through α3/α5/β4* nAChRs, which are only activated in the presence of high nicotine concentrations (2 mg/kg in a rodent) [47,106,107]. These nAChR subunits are highly populated in the habenulo-IPN pathway that mediates the aversive properties of nicotine. In the absence of these subunits, individuals no longer titrate their nicotine consumption and lose the mediatory effects of this pathway on the negative effects of high doses of nicotine. These subunits are also important in the manifestation of withdrawal, including the somatic and physical symptoms. Knockout mice lacking these subunits exhibited fewer somatic withdrawal symptoms [150,151] in chronic nicotine administered mice.

Further, the less studied but still relevant aversive LHb pathway is a collection of glutamatergic and cholinergic excitatory neurons that project to the tail of the VTA, the rostromedial tegmental nucleus (RMTg). The GABAergic neurons of the RMTg then inhibit DA cell firing of the VTA [152]. This pathway is acted on by aversive stimuli, such as the aversive signals associated with nicotine withdrawal and the absence of an expected reward, in an attempt to suppress the mesocorticolimbic DA system [153]. Activation of the Hb pathways is important in the development of addiction, for their influence on drug-seeking and taking behaviors. 

## 4. Flavoring Chemicals in Nicotine Addiction

In the late 1900s, cigarette smoking had little to no restrictions and was acceptable in restaurants, hospitals, public transportation, and more, making them not only freely accessible but a new trend among the United States, with 50% of young women and 60–70% of young men being long-term smokers [154]. This popularity was in part attributed to flavoring additives. Mentholated cigarettes became a huge success, especially among the adolescent population after the discovery that menthol provided a cooling sensation in the cigarette smoke. Adolescent cigarette use continued to rise over the years due to the sweet aroma and taste that “characterizing” flavors provided. It was not until 2009 that the Family Smoking Prevention and Tobacco Control Act (FSPTCA) gave the FDA the authority to ban “characterizing” flavor additives such as strawberry and vanilla (not including tobacco or menthol) in combustible cigarettes in an attempt to mitigate adolescent cigarette use [155].

Since the FSPTCA, menthol has been the most widely studied flavor additive, for its popularity and sole acceptance as a flavorant in combustible cigarettes. Aside from mentholated cigarette popularity among adolescents, menthol has also long been popular among African Americans, with more than 80% of non-Hispanic black adults using menthol cigarettes [156]. It was recently determined that African Americans exhibit unique variants that promote a significant increase in the odds of menthol cigarette smoking [157]. Now that ENDS products are commonly used among those that were life-long smokers, menthol is still a very prominent flavor in e-liquids. Menthol is known to intensify various smoking-related behaviors and facilitates first-time nicotine use by masking the aversive properties of nicotine [158], which may explain the popularity of flavored ENDS and signifies the concerns with adolescent ENDS use [24,159].

Initially, menthol was considered to be an inert flavor additive, with the focus of tobacco-related research being on nicotine dependence. It was not until the early 2000s that studies on menthol began to surface. These initial studies suggested menthol smokers to be more nicotine dependent based on (1) how quickly they smoke upon waking in the morning compared to non-menthol smokers [160,161] and (2) the higher nicotine/cotinine levels among menthol smokers versus non-menthol smokers [161,162]. Further, Ahijevych and Garrett identified menthol to be a conditioned stimulus that enhances the rewarding and reinforcing properties of nicotine through its positive sensory effects [162], resulting in drug-craving and drug-taking behaviors. These behaviors were mimicked in female rats that earned more intravenous nicotine infusions in the presence of an oral menthol cue during self-administration assays [16]. Conditioned stimuli paired with nicotine enhance the acquisition and maintenance of nicotine use and are known to be a driving force in nicotine dependence [163]. Menthol has the unique minty taste profile and a cooling sensory effect that, when combined with nicotine can contribute to craving and relapse when a person is going through an abstinent period [164]. These sensory cues are common among other sweet oral flavorants as well, including sucrose and saccharin. Interestingly, Wickham et al. utilized an intraoral delivery method that resulted in sucrose- and saccharin-induced phasic DA release, and significantly increased nicotine self-administration behavior [165]. It is these conditioned sensory cues that often make cessation so difficult.

Based on these initial sensory effects, menthol was studied for its pharmacological impact in the brain. Menthol was determined to act as a negative allosteric modulator (NAM) of nAChRs following the observation of reduced nicotine-induced (but not ACh-induced) inward currents through α4β2* nAChRs expressed in human embryonic kidney (HEK) cells [166]. This finding was supported by additional electrophysiological characterization utilizing single-channel recordings that revealed menthol shifted α4β2 nAChRs towards the desensitized conformation state. In separate investigations, menthol was also determined to act as a noncompetitive antagonist on both α7 and α3β4* nAChRs [167,168]. A combination of computational docking, site-directed mutagenesis, and electrophysiology revealed that menthol may exert its effect on nAChRs by binding to the 9′ position in the transmembrane 2 (M2) helix of nAChRs [22] (Figure 4A). Here, a series of mutations at the 9′ leucine residues (L9′) of the high-sensitivity α4β2 nAChRs were made and characterized with electrophysiology to investigate the connection between menthol’s pharmacology and the L9′ site [22]. It was concluded that menthol’s inhibitory actions rely on the presence of the L9′ site of the M2 helix and only one menthol molecule is sufficient for α4β2 nAChR inhibition. Another recent study identified additional menthol binding sites, including the M3-M4 extracellular interface on α4 subunits, various sites at the M1–M4 interface on β2 subunits, and M2 residues positioned extracellularly [169]. However, these binding sites have not been validated using functional assays. Here, it is important to distinguish the fact that the concentrations of menthol used to examine its inhibitory activity on nAChRs is several orders of magnitude higher (>30 µM) than what is considered smoking/vaping-relevant (<2 µM). The studies that have used low concentrations of menthol (0.5 µM) have determined that menthol combined with nicotine does enhance nAChR upregulation on VTA dopamine neurons and also enhances the excitability of these same neurons (discussed further below) [18,22]. A previously published review has extensively disseminated the recent investigations that have elucidated the mechanism of menthol’s actions on nicotine reward and reinforcement [158]. To avoid repeating what has been eloquently explained and summarized previously, we direct interest into the specifics to the review by Wickham et al. [158].

Much like nicotine’s prominent ability to promote nAChR upregulation, menthol was determined to induce nAChR upregulation in the brainstem, cerebellum, corpus callosum, PFC [21], hippocampus, striatum [170], and VTA [18,20]. More specifically, Henderson and colleagues observed increased levels of α4 and α6 nAChR subunit expression in VTA neurons following 10-day chronic menthol exposure [20], as well as increased levels of α4 and α4α6* nAChRs in VTA neurons following 10-day chronic menthol + nicotine exposure [18] through osmotic minipumps and daily intraperitoneal injections in mice. It is important to once again distinguish the concentrations used in these studies [18,20]. However, unlike nicotine, chronic menthol treatment caused a shift from α4_(2)_β2_(3)_ (high-sensitivity) to α4_(3)_β2_(2)_ (low-sensitivity) nAChRs in cultured cells (Figure 4B) [20]. This stoichiometry shift was accompanied by reduced VTA DA neuron firing frequency and an attenuation of reward-related behavior, dissimilar to nicotine. Interestingly, menthol + nicotine not only resulted in α4α6* nAChR upregulation, but also enhanced VTA DA neuron excitability, and enhanced nicotine reward-related behavior [18].

These neurobiological and neurophysiological alterations via menthol contribute to the behavioral effects demonstrated in rodent studies. As stated previously, menthol lacks the rewarding properties that nicotine presents, but menthol has the ability to enhance nicotine reward-related behavior with rodents in a conditioned place preference assay [18]. Further, to determine menthol’s effect on smoking initiation, self-administration paradigms have been used to determine menthol’s role in nicotine acquisition. In numerous studies, menthol facilitated intravenous nicotine self-administration and increased the rate of nicotine intake compared to control groups [16,17]. This behavior was followed by enhanced withdrawal-like behaviors, demonstrated through somatic signs and anxiety-related assays [170], and enhanced menthol-induced reinstatement [16]. Menthol presents a minty and cooling sensation through a transient receptor potential M8 (TRPM8)-mediated mechanism, which in turn, masks the aversive and harsh taste of nicotine by shifting nicotine’s inverted U-shaped dose-response curve to the left and contributing to the enhanced nicotine acquisition behavior. Aside from menthol’s ability to enhance smoking initiation, chronic menthol enhances the reinforcing properties of nicotine, leading to enhanced nicotine self-administration and dependence over time [17]. This may partially be due to menthol’s effect on nicotine metabolism [171]. More specifically, menthol smokers exhibit reduced nicotine metabolism compared to non-menthol smokers [170,172], likely due to a competitive effect of menthol on the availability of CYP2A6, the major cytochrome enzyme involved in metabolizing nicotine [171].

With the common knowledge that smokers of menthol cigarettes exhibit lower cessation rates [173] and the growing use of flavored ENDS products (see Table 2), studies are beginning to arise on the role of other flavor additives. Recently these studies have included the popular green apple ENDS flavorants, farnesol [26] and farnesene [27], for they are structurally similar to menthol in the terpene class (see Figure 5). According to Espino-Diaz et al., apple biochemistry varies during the maturation process, with aldehydes primarily making up the apple flavor profile during the beginning, and following maturity, the flavor profiles transition to primarily alcohols and esters [25]. Farnesol, like menthol, is a terpene alcohol, which makes them more water soluble than a typical hydrocarbon, given their hydroxyl functional group. Both farnesene and farnesol differ from menthol given they are acyclical sesquiterpenes (natural 15-carbon organic compounds consisting of three isoprene units). Sesquiterpenes, alongside monoterpenes, are the main aroma components that make up apple flavorants. Additional to these two flavorants, green apple flavor includes numerous other flavorant compounds including hexyl acetate, ethyl acetate, and methylbutyl acetate [174]. However, ENDS-related studies on these flavorants have not yet been performed.

Similar to menthol, both farnesol and farnesene exhibited a significant enhancement of nicotine reward-related behavior in mice [26,27]. In the case of farnesol, there was an observed sex-dependent effect as only male mice (at the doses tested) exhibited changes in reward-related behavior. Interestingly, unlike menthol, both green apple flavorants exhibited significant reward-related behavior in the absence of nicotine compared to saline-control. The finding that ENDS flavors may produce reward-related behavior on their own is a significant contribution to the field of nicotine addiction and may explain the continued use by adolescents and their strong preference for fruity flavors. This also reveals some insight into why there may be a rise in vaping of zero-nicotine flavored e-liquids.

Much like the previous menthol findings, Avelar et al., identified farnesol-induced upregulation of α6* and α4α6* nAChRs on VTA DA neurons and a complementary increase in the firing frequency of these neurons [26]. Here, it was reported that farnesene, by itself, produced a greater change in nAChR upregulation and enhancement in firing frequency when compared to farnesol plus nicotine. They speculated that the differences in the flavorant-induced reward is likely due to the fact that menthol-alone only upregulates low-sensitivity α6* (not α4α6*) nAChRs and decreases VTA DA neuron firing [20], whereas farnesol not only increased VTA DA neuron firing but also upregulated both α6* and α4α6* nAChRs [26]. Desensitization of the α4β2* nAChRs on SNr GABA neurons reduces the inhibitory outputs, and results in disinhibiting the VTA DA neurons [175]. Based on these previous findings, farnesol treatment was observed on SNr GABA neurons for their potential downstream effects on VTA DA neurons [26]. Here, in male mice only, they found a significant downregulation of α4 subunits on SNr GABA neurons and this contributed to a decrease in the firing frequency of these neurons.

Similar to the farnesol study, Cooper et al. performed conditioned place preference assays with mice and observed significant farnesene-induced reward-related behavior [27]. Unlike farnesol, farnesene exerted its effects through changes in nAChR stoichiometry, not nAChR upregulation. Through the use of mouse brain slices and cultured neuroblastoma 2a cells, it was determined that farnesene triggered a stoichiometric shift toward high-sensitivity α4β2 nAChRs on VTA DA neurons. This was accompanied by a leftward shift in the concentration-response of nicotine-induced inward currents on VTA DA neurons in a brain slice preparation. Finally, VTA DA neurons in farnesene-treated mice exhibited an increase in excitatory postsynaptic current frequency and amplitude. Considering that this finding was the result of a coronal brain-slice preparation, this suggests farnesene alters local VTA GABAergic or glutamatergic transmission and results in a net increase of excitatory input on to VTA DA neurons. The above flavorant studies demonstrate the potential risk for the abuse potential of ENDS flavorants and highlight the need to understand zero-nicotine flavored products.

## 5. Future Directions

Although flavorant studies are becoming more prevalent, there is still much to be learned. Currently, menthol and green apple are the only flavors studied for their effects on nicotine addiction-related behaviors. Given the distinct differences between menthol and green apple, and more specifically between green apple flavorants: farnesol and farnesene, it is safe to assume that the various flavor profiles on the market will have differing effects in future studies as well. This is especially true given that prominent chemical flavorants used in ENDS e-liquids have similar chemical scaffolds as menthol, farnesene, and farnesol (see Figure 5). It is likely these effects will also differ among specific classes of flavor chemicals being studied. For example, majority of the flavorants used in these devices are alcohols or aldehydes, and some flavorants are present in numerous flavor subcategories outside of their major flavor category (i.e., vanillin (vanilla) flavorant in chocolate, mint, and cotton candy flavored products; [174]). It will be interesting to determine how heavily used flavorants (like vanillin) differ from farnesol or farnesene, which are primarily found in apple flavored products. As well as general differences in chemical classes.

Another important gap in this field of research is the gap in knowledge regarding the difference between adult, adolescent, and in utero exposure. This is a critical gap to fill considering the constant rise in adolescent ENDS use. Despite not knowing the age-dependent effects on flavored ENDS use, it is common knowledge that adolescent nicotine use has a variety of detrimental impacts on the developing brain including greater tolerance for high doses of nicotine [176], enhanced nicotine reward sensitivity [177], reduced PFC activation and thus impaired cognitive function [178,179], and enhanced risk for other drug abuse later in life through a process known as “priming” [146,180]. With the current flavor studies resulting in an enhancing effect on nicotine pharmacology, it is likely flavors may also enhance the above neurological impacts as well. Yet, more information must be gathered.

Currently, menthol and green apple have been studied for their effects on midbrain dopamine and GABA neurons, however glutamatergic and cholinergic afferents are important contributors to the activity of the mesocorticolimbic pathway. It would be interesting to determine the role that flavors may play in glutamate firing, including synaptic plasticity. It is also critical to utilize fast-scan cyclic voltammetry in the determining of flavorant-induced DA release in the NAc. Although we have observed flavorant-induced alterations in firing on presynaptic midbrain neurons, it is important to determine the postsynaptic effects as well.

With crucial evidence pointing to the aversive pathways as a mediator for nicotine addiction, it is also important to determine if flavors also elicit effects in these specific regions. Flavorants may be acting on these pathways by reducing the inhibitory transmission from the RMTg and IPN to the VTA, resulting in net excitation and dopamine release. Furthermore, the habenula consists of more diverse nAChR subtypes than the midbrain. Flavorants may have a more (or less) pronounced effect on these subtypes. Regardless, there is plenty of research that still needs to be done in order to fully understand how these devices and their additives affect a chronic user.

Lastly, although combustible cigarette plus electronic cigarette use (termed ‘dual-use’) has been explored. These studies have focused more on lung-related and cardiovascular-related impacts, as opposed to the impacts on the brain. It will be interesting to identify the effect that dual use has on a chronic user’s brain and whether ENDS use exacerbates the already-addicted brain, or merely maintains the combustible cigarette-induced neuroalterations.

## 6. Conclusions

Here, we discussed the major flavorant-induced changes in neurobiology and neurophysiology that include changes in nAChR function, nAChR subtype assembly, and nAChR upregulation. Together, these changes cause an impact on reward- and reinforcement-related behavior. Thus far, basic science investigations into smoking- and vaping-related behavior have only focused on menthol and green apple flavorants. These studies suggest that there are complex effects that ENDS flavorants exert on nicotine reward and reinforcement. Based on these findings, and the current knowledge regarding nicotine, it is important to continue investigating ENDS flavorants to determine their impact on the neurobiology and neurochemistry of addiction. In parallel, there is a need to understand the toxicology of these ENDS flavorants as well.

## Figures and Tables

**Figure 1 molecules-25-04223-f001:**
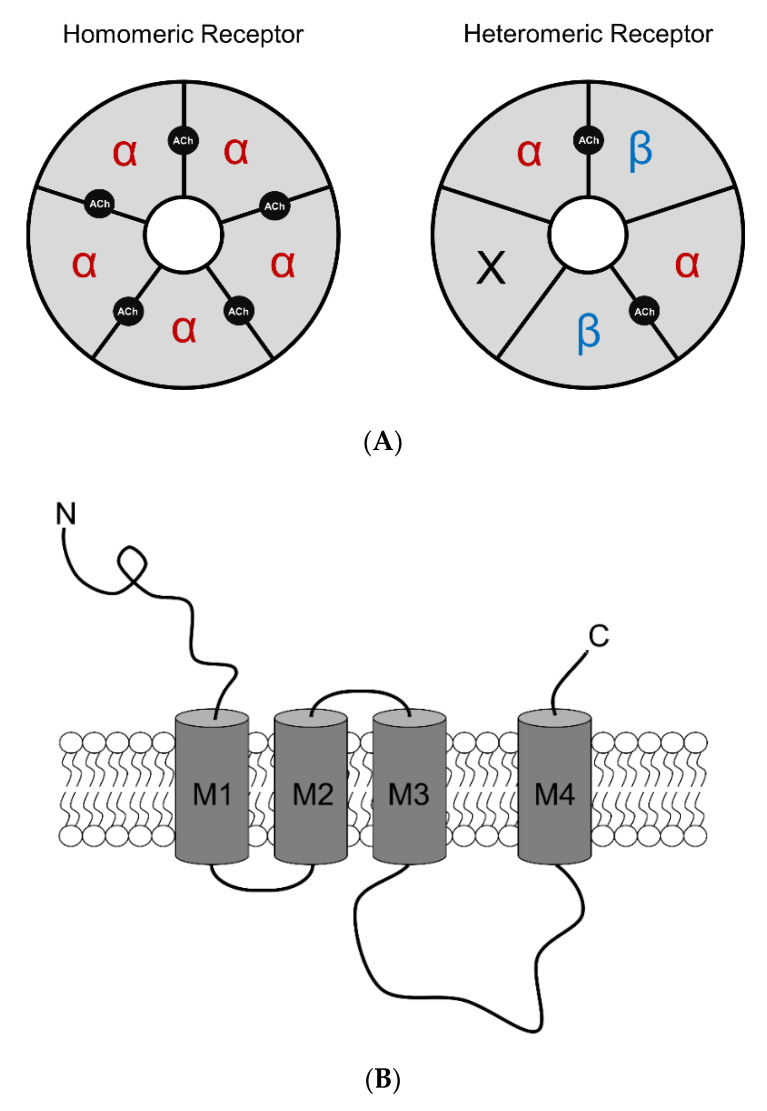
Human nAChR structure and assembly. (**A**) Homomeric and heteromeric nAChR complexes assemble as solely α-subunits or α/β-subunits, respectively. Homomeric nAChRs possess five agonist binding sites at each α-α interface while heteromeric nAChRs possess two agonist binding sites at the α-β interface but can still be weakly activated by ‘non-canonical’ sites at the α-α interface (if present). X indicates other subunits may be present. (**B**) Single nAChR subunit topology consists of an extracellular domain, four transmembrane domains, and an intracellular loop that varies in length depending on subtype. (**C**) Pentameric nAChR complex from a top-side view identifying the formation of the transmembrane domains of individual nAChR subunits with respect to the nAChR central pore.

**Figure 2 molecules-25-04223-f002:**
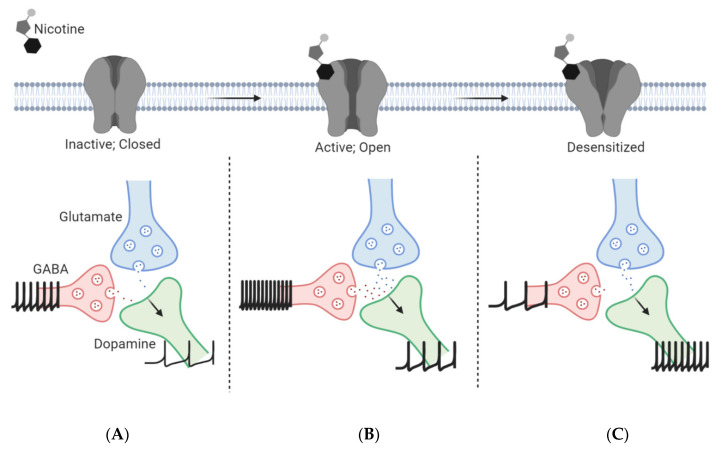
Involvement of ventral tegmental area (VTA) neuron types in nicotine reward and reinforcement. (**A**) In the absence of nicotine, glutamate and GABA inputs to VTA dopamine neurons modulate activity of the mesolimbic reward pathway. (**B**) Acute nicotine on α7 and α4β2 nAChRs elicits enhanced glutamatergic and GABAergic firing, respectively, onto VTA dopamine neurons, resulting in a net enhancement of dopaminergic neuron firing and subsequent dopamine release. (**C**) Following long-term nicotine use, α4β2 nAChRs desensitize rapidly with acute exposure to nicotine and result in reduced GABAergic firing while glutamatergic firing is enhanced to stimulate burst firing of dopaminergic neurons (see also Table 1 for additional details).

**Figure 3 molecules-25-04223-f003:**
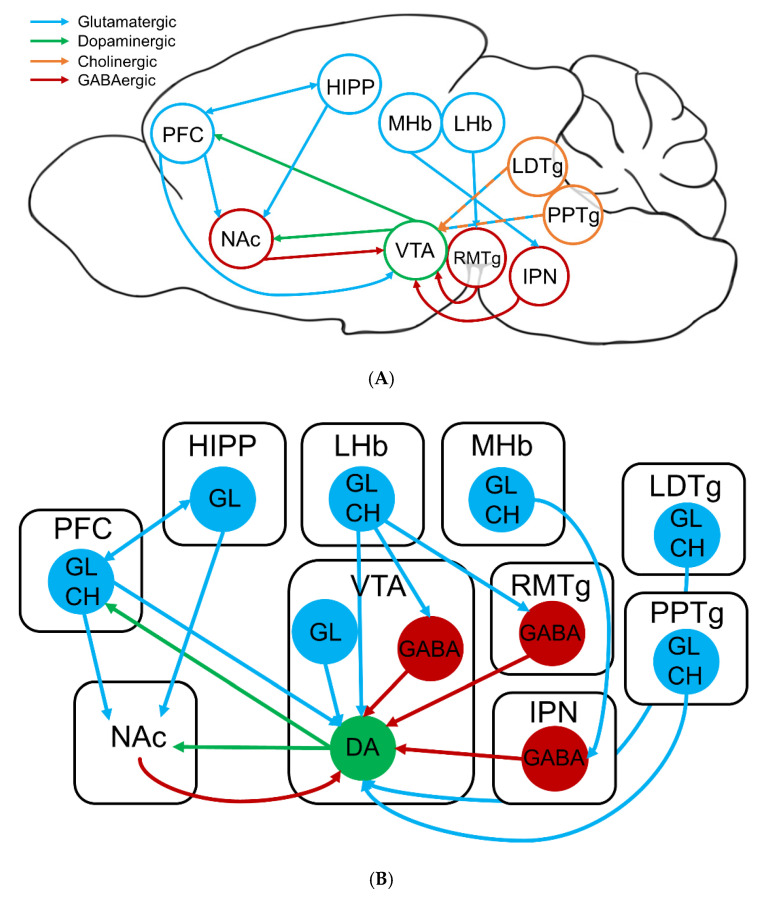
Neurocircuitry involved in nicotine addiction. (**A**) Sagittal mouse brain schematic of the major brain regions and connections involved in nicotine addiction. (**B**) Intricate schematic of the above neurocircuitry. DA: dopaminergic, GABA: GABAergic, GL: glutamatergic, CH: cholinergic neurons. Blue arrows indicate excitatory projections; red arrows indicate inhibitory projections; and green arrows indicate modulatory projections.

**Figure 4 molecules-25-04223-f004:**
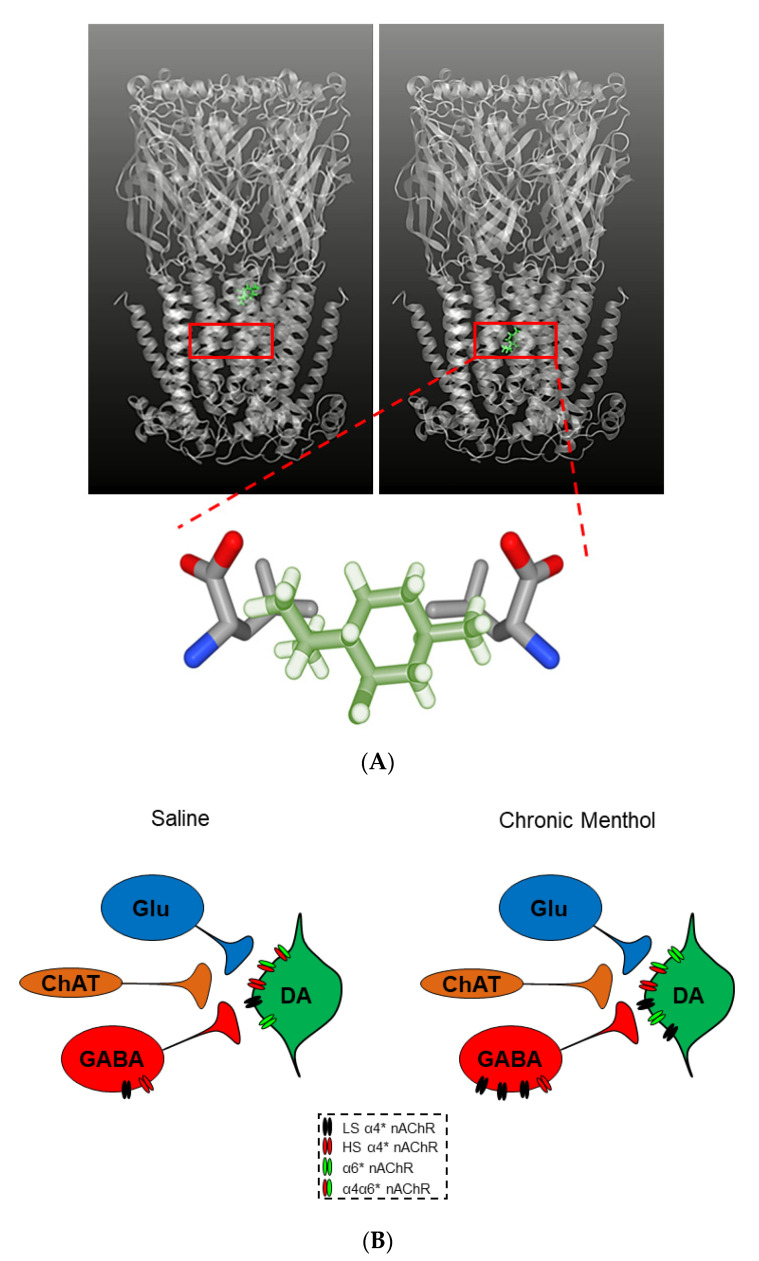
Menthol binding patterns and mechanisms of action. (**A**) Top panel: Menthol molecule (in green) binding in the transmembrane 2 (M2) helix lining the central pore of the nAChR. Bottom panel: A closer look at menthol binding within the 9′ leucine residues of the M2 helix of the nAChR. (**B**) Compared to saline, menthol induces upregulation of low-sensitivity (LS) α4* nAChRs on VTA GABA neurons (in red) and α4* (LS) and α6* nAChRs on VTA dopamine neurons (in green). This results in reduced neuronal firing based on the desensitization properties of the nAChRs, with a net excitatory effect of VTA dopamine release. * = nAChR may contain other subunits.

**Figure 5 molecules-25-04223-f005:**
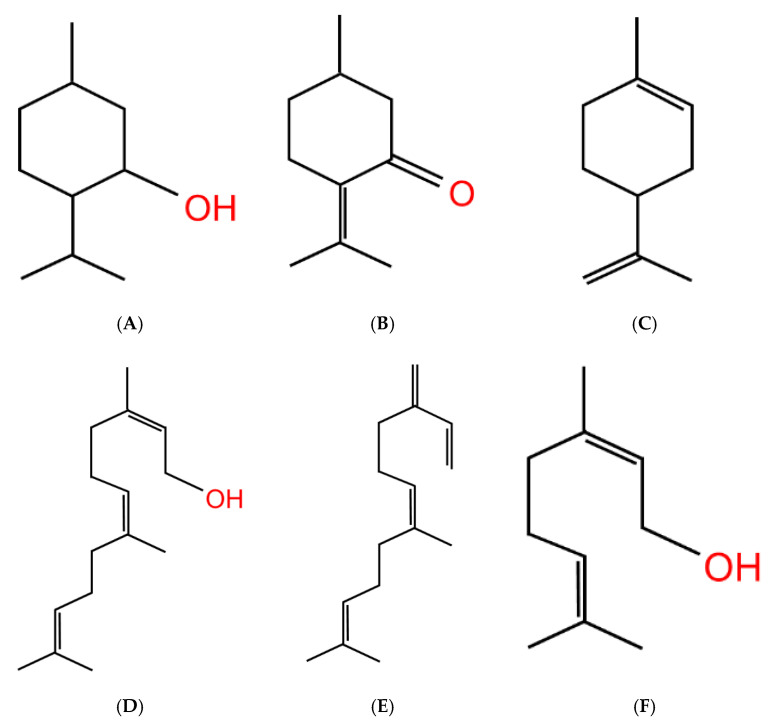
Flavorant chemical profiles. Similarity in chemical structure formation of (**A**) menthol, (**B**) pulegone (peppermint), (**C**) limonene (citrus), (**D**) farnesol, (**E**) farnesene, and (**F**) geraniol (fruity), despite the wide array of flavor profiles.

**Table 1 molecules-25-04223-t001:** Nicotine’s actions in the brain.

Steps of Nicotine Addiction	nAChR Subunits	Brain Regions	Nicotine Concentration	Duration of Administration
Nicotine binding promotes a conformational transition of nAChRs from a resting, closed state to an open state, allowing signal transduction to occur [65,66].	α4β2*, α4α6β2*, α6β2β3, α7	Midbrain	Small (<1 µM)	Acute
When an individual becomes a long-term nicotine user, nAChR upregulation occurs.	α4, α6, β2, and β3	VTA	20–500 nM	Chronic
Upregulated α4β2* nAChRs found on GABA neurons of the midbrain elicit an inhibitory tone to VTA DA neurons, reducing the transmission of DA and contribute to nicotine-seeking behaviors [72].	α4β2*	Midbrain GABA		
After long-term exposure to nicotine, α4β2* nAChRs desensitize quickly, which decreases the GABAergic transmission onto DA neurons resulting in the disinhibition of DA neurons.	α4β2*	Midbrain GABA	20–500 nM	Chronic
The activation of α4*, α6*, and α7 nAChRs on midbrain DA and glutamate neurons, respectively, promotes a net excitatory effect on DA neurotransmission from the VTA to the NAc and PFC, leading to the rewarding and reinforcing aspects of nicotine use [71].	α4β2*, α6β2β3, α4α6β2*, α7	VTA		
These effects are reinforced by the enhanced glutamate neurotransmission from distal regions and local mVTA glutamate neurons to lateral VTA (latVTA) DA neurons [63,73], leading to long-term potentiation (LTP).	α7	mVTA, PFC		

Summarized details of the various steps (sensitization, upregulation, and desensitization) that take place during the development of nicotine dependence. The major findings for each step are indicated in the leftmost column. The specific nAChR subunits and brain region associated with these changes are in the second and third column. The nicotine concentration and duration of administration that induce these changes are located in the two rightmost columns. * = nAChR may contain other subunits.

**Table 2 molecules-25-04223-t002:** Popular flavoring chemicals in e-liquids.

Flavor Chemical	Chemical Class	Flavor Profile
Vanillin	Aldehyde	Vanilla, Chocolate, Cotton Candy, Mint, Coffee, Tobacco
Ethyl Vanillin	Aldehyde	Vanilla, Chocolate, Cotton Candy, Coffee, Tobacco
Ethyl Maltol	Alcohol	Vanilla, Chocolate, Cotton Candy, Mint, Coffee, Tobacco, Grape, Cherry
Maltol	Alcohol	Vanilla, Chocolate, Mint, Coffee, Tobacco, Grape
Benzaldehyde	Aldehyde	Cherry, Bubble Gum
Benzyl Alcohol	Alcohol	Cherry, Vanilla, Coffee, Tobacco
Ethyl Butyrate	Ester	Vanilla, Cherry, Bubble Gum, Apple, Tobacco, Grape
Menthol	Alcohol	Mint
Hexyl Acetate	Ester	Apple
Ethyl Acetate	Ester	Bubble Gum, Apple, Grape, Tobacco
Methylbutyl Acetate	Ester	Bubble Gum, Apple
Farnesol	Sesquiterpene	Apple
Farnesene	Sesquiterpene	Apple

Comprehensive list of popular flavoring chemicals including their chemical class association and various flavor profiles that varying concentrations of these flavoring chemicals can be found in.

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
