# Peer review of "The Impact of Electronic Nicotine Delivery System (ENDS) Flavors on Nicotinic Acetylcholine Receptors and Nicotine Addiction-Related Behaviors"

_molecules, 2020, doi:10.3390/molecules25184223_

Round 1

Reviewer 1 Report

In the review of Cooper and Henderson entitled “A new generation of nicotine users: the impact of electronic nicotine delivery system (ENDS) flavors on nicotine addiction” they discussed the major flavorant-induced changes in neurobiology and neurophysiology that include changes in nAChR function, nAChR subtype assembly, and nAChR upregulation.

The review summarizes the current knowledge on nicotine addiction and the various brain regions and nicotinic acetylcholine receptor subtypes involved. Also, it describes the most recent findings regarding flavorants and their roles in nicotine addiction and vaping-related behaviors.

The authors have worked on a well-written and exhausting review. They conclude that it is important to continue investigating ENDS flavorants to determine their impact on the neurobiology and neurochemistry of addiction, and that there is a need to understand the toxicology of these ENDS flavorants.

In my opinion, overall, this is a good review with balanced assessment of the role of ENDs flavorants in nicotine addiction.

The topic of the manuscript is of some importance. The manuscript is well written and presented in a logic manner.

Perhaps, Line: 67, “…of nicotine in an attempt to combat the growing ENDS-use epidemic.” I think that by using the word “epidemic” makes a “strong” sentence that parallels to the COVID-19 situation, whereas ENDs are not proven, yet so deadly!

To conclude, my overall opinion about this manuscript is that it can be considered for publication, upon journal editor’s priorities.

Author Response

We would like to thank Reviewer 1 for their feedback on our manuscript.

Reviewer 2 Report

The authors Cooper and Henderson reviewed the current knowledge on the role of nAChRs in nicotine addiction and mechanisms via which tobacco flavor additives, menthol and green apple flavorants (farnesol and farnesene), enhances the effect of nicotine. The summary was succinct and up-to-date. A few minor suggestions are provided below for authors' consideration:

1. A large portion (almost half) of the text was devoted to the role of nAChRs on nicotine addiction. This is section is well written but was not given credit in the title and abstract section. Therefore, it was a pleasant surprise to read this section. It would be better if the title and the abstract give the reader some hints about this section. It can further increase the audience of this review.

2. Green Apple flavorant studies are very new. It will be better to include the green apple in the abstract to draw more readership.

3. Figure 2. It is not clear what the key message is for this figure. Which nAChR subtype are these, where are they located on each cell type? Does desensitization cause compensatory upregulation?

4. Title of Section 2.3. "Neuronal nAChR subtypes". The n in nAChR is Neuronal. There are several other places using the same repetitive nomenclature.

5. Genetics studies have identified the role of most nAChRs in smoking, not only alpha3, alpha5, which the paper cited. A key report is PMID: 30643251

6. There is also a genetic study on menthol cigarette smoking https://doi.org/10.1371/journal.pgen.1007916

7. Menthol also affects nicotine metabolism. This body of literature is ignored in the manuscript.

8. Line 257. "Although nicotine reward prompts a euphoric sensation upon acute administration," This statement needs citation. I am aware of several papers that state the opposite. PMID: 28190083 PMID: 17235611.

9. Figure 3B. What are GL and CH? I have my guesses, but you better tell me.

Reviewer 3 Report

The current manuscript provides a comprehensive literature review regarding the nicotine and flavors mediated addiction and it is implication to ENDs. This manuscript addresses a critical question, that is needed to increase public awareness about the potential harm associated with these products and the urgency for further evidence-based effects of ENDS on brain. While the manuscript provides comprehensive review of literature in a well written manner, several issues need to be addressed:

  • Adding prevalence data from 2019 NYTS will be more attractive as it shows a huge increase in ENDs usage especially among youth.
  • In the context of nicotine addiction, authors should consider discussing the nicotine concentration and kinetics differences between traditional smoking and ENDs including newer designs such as JUUL (most commonly used).
  • While section 2.2 provides a comprehensive review of nicotine actions in the brain, it is hard to follow. The manuscript would benefit from adding a table summarizing these findings including: receptors subtypes involved, brain region affected, nicotine concentration, and duration of administration (acute vs. chronic), etc.
  • It may be useful to discuss the role of glycine and glycinergic receptor in addiction as their roles are emerging and many have shown that glycinergic synapses are also present in the VTA and are involved in substance and drug abuse and dopaminergic communication.
  • Adding a figure showing the proposed mechanism of menthol mediated nicotine effect will improve the manuscript significantly.
  • While mentioned briefly, the authors need to provide a comprehensive list of commonly used flavors in the market to provide a basis for future neurobiological studies. Also, discussing the impact of the sugar-based flavors on addition and craving.
  • It would be interesting to discuss the dual use of e-cig and smoking and it is impact on brain.
  • Figures legends are brief and not descriptive.

Round 2

Reviewer 3 Report

The revised manuscript improved significanly and authors addressed all my comments.